

# The *ddeq* Python library for point source quantification from remote sensing images (Version 1.0)

Gerrit Kuhlmann[1], Erik F. M. Koene[1], Sandro Meier[1,2], Diego Santaren[3], Grégoire Broquet[3], Frédéric Chevallier[3], Janne Hakkarainen[4], Janne Nurmela[4], Laia Amorós[4], Johanna Tamminen[4], and Dominik Brunner[1]

[1]Swiss Federal Laboratories for Materials Science and Technology (Empa), Dübendorf, Switzerland
[2]Remote Sensing Laboratories, University of Zurich, Switzerland
[3]Laboratoire des Sciences du Climat et de l'Environment, LSCE/IPSL, CEA-CNRS-UVSQ, Université Paris-Saclay, Gif-sur-Yvette, France
[4]Earth Observation Research, Finnish Meteorological Institute (FMI), Helsinki, Finland

**Correspondence:** Gerrit Kuhlmann (gerrit.kuhlmann@empa.ch)

**Abstract.** Anthropogenic emissions from "hotspots", i.e. cites, power plants and industrial facilities, can be determined from remote sensing images obtained from airborne and space-based imaging spectrometers. In this paper, we present a Python library for data-driven emission quantification (*ddeq*) that implements various computationally light methods such as Gaussian plume inversion, cross sectional flux method, integrated mass enhancement method and divergence method. The library provides a shared interface for data input and output as well as tools for pre- and post-processing of data. The shared interface makes it possible to easily compare and benchmark the different methods. The paper describes the theoretical basis of the different emission quantification methods and their implementation in the *ddeq* library. The application of the methods is demonstrated using Jupyter Notebooks included in the library, for example, for $NO_2$ images from the Sentinel-5P/TROPOMI satellite and for synthetic $CO_2$ and $NO_2$ images from the Copernicus $CO_2$ Monitoring (CO2M) satellite constellation. The library can be easily extended for new datasets and methods, providing a powerful community tool for users and developers interested in emission monitoring using remote sensing images.

## 1   Introduction

The majority of anthropogenic emissions of air pollutants and greenhouse gases is confined to localized sources such as cities, power plants and industrial facilities (e.g., Crippa et al., 2022). The emissions of these "hotspots" can be determined from the atmospheric plumes of trace gas column densities in remote sensing images. Trace gases of interest are nitrogen dioxide ($NO_x = NO_2 + NO$), carbon monoxide (CO), carbon dioxide ($CO_2$), methane ($CH_4$) and others. Numerous methods have been developed in recent years to quantify the emissions from remote sensing images by matching observations to simulated plumes (e.g., Bovensmann et al., 2010; Nassar et al., 2017; Broquet et al., 2018; Ye et al., 2020; Lei et al., 2021; Kaminski et al., 2022), applying the principle of mass conservation to individual or temporally averaged images (e.g., Beirle et al., 2011; de Foy et al., 2015; Varon et al., 2018; Reuter et al., 2019; Zheng et al., 2020; Kuhlmann et al., 2021; Leguijt et al., 2023), or





using machine-learning models trained with synthetic observations (e.g., Jongaramrungruang et al., 2022; Joyce et al., 2023; Dumont Le Brazidec et al., 2023b).

Remotely sensed trace gas images are available from space-based imaging spectrometers such as the Tropospheric Monitoring Instrument (TROPOMI, Veefkind et al., 2012) and the Orbiting Carbon Observatory-3 (OCO-3, Eldering et al., 2019), as well as from high-resolution point source imagers such GHGSat, several hyperspectral land imaging sensors (e.g., Sentinel-2), and from the upcoming generation of missions in polar and geostationary orbits (i.e. CO2M, GEMS, GOSAT-GW, Sentinel-4 and -5, TEMPO and others). In addition, airborne imaging spectrometers can be used to map emission plumes at high spatial resolution (e.g., Thorpe et al., 2017; Tack et al., 2019; Fujinawa et al., 2021). An example of the plumes that can be visible from a city or power plant is shown in Figure 1. This example is from the synthetic SMARTCARB dataset (Kuhlmann et al., 2019) generated for the Copernicus $CO_2$ Monitoring (CO2M) satellite constellation planned for launch in 2026 (ESA Earth and Mission Science Division, 2020).

Measurement-based emissions monitoring systems are currently being developed to support global efforts on reducing greenhouse gas emissions to achieve the goals of the Paris Agreement for Climate Change. One such system is the European $CO_2$ Monitoring and Verification Support (CO2MVS) capacity that will be implemented as part of the European Copernicus programme (Janssens-Maenhout et al., 2020). A prototype system of the European CO2MVS capacity is build in CoCO2 project (https://coco2-project.eu/). Since one goal of CO2MVS is the global monitoring of emission hotspots, several emission quantification methods were implemented and benchmarked in the CoCO2 project using synthetic CO2M $CO_2$ and $NO_2$ observations and Sentinel-5P/TROPOMI $NO_2$ observations (cf., Hakkarainen et al., 2023; Santaren et al., 2023). As the new generation of satellites will provide a large number of hotspot images (e.g., Kuhlmann et al., 2021; Wang et al., 2020), it is foreseen that computationally lightweight methods will be needed to process the large amount of data in the operational CO2MVS system. Such methods will have to make optimal use of the information contained in the images without requiring expensive plume simulations with a high-resolution atmospheric transport model.

This paper describes the Python library for data-driven emission quantification (*ddeq*; Version 1.0), developed in the CoCO2 project as a shared library for the implementation of various lightweight approaches. Although the various methods differ in many aspects, they share many pre- and post-processing steps including data input and output. The common interface thus makes *ddeq* a powerful tool for comparing and benchmarking different methods but also for implementing new approaches. The *ddeq* library was originally developed in the SMARTCARB project for detecting and quantifying $CO_2$ and $NO_x$ emissions in synthetic $CO_2$ and $NO_2$ images of the CO2M mission (Kuhlmann et al., 2019, 2020a, 2021), but it has also been used for quantifying $NO_x$ emissions with the airborne APEX imaging spectrometer (Kuhlmann et al., 2022). Whereas *ddeq* is designed for the lightweight emission quantification of hotspots from remotely sensed images, a similar community-driven library exists for regional and global emission estimation with atmospheric model runs through the Community Inversion Framework (CIF, Berchet et al., 2021). The *ddeq* version presented here does not include machine-learning models, which were also considered in the CoCO2 project (Dumont Le Brazidec et al., 2023a, b) but they were not included here. *ddeq* has been used in the CoCO2 project for benchmarking the different methods using synthetic CO2M observations (Santaren et al., 2023) and TROPOMI $NO_2$ observations (Hakkarainen et al., 2023).



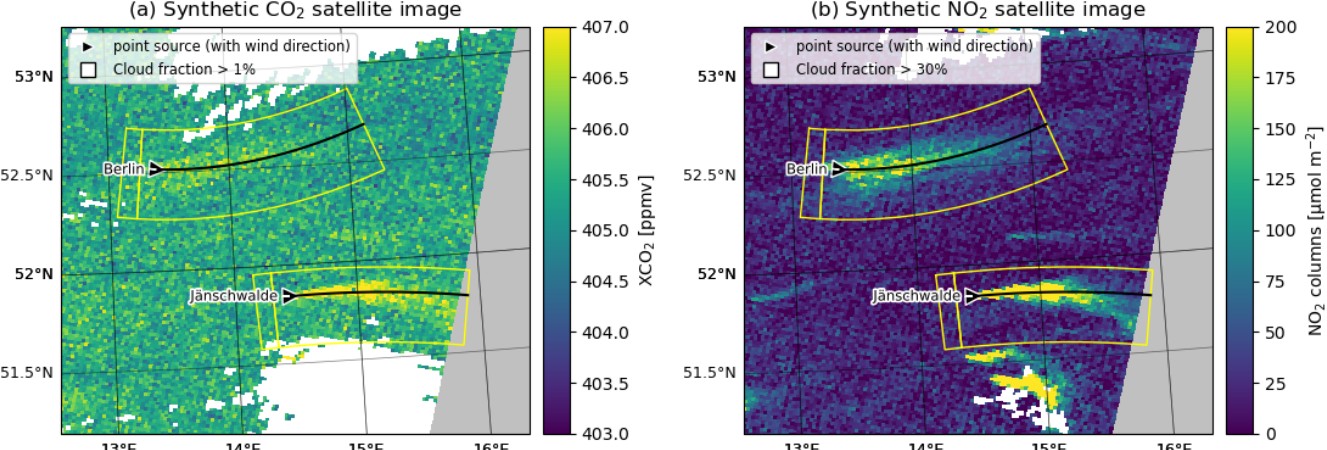

**Figure 1.** Example of synthetic CO2M (a) $CO_2$ and (b) $NO_2$ satellite images from the SMARTCARB dataset. The images show the emission plumes of the city of Berlin and the coal-fired power plant near Jänschwalde at 2 km resolution. Pixels with cloud fractions larger than 1% for $CO_2$ and 30% for $NO_2$ are shown in white and regions outside the satellite swath are shown in gray. The triangular marker indicates the wind direction at the source. The yellow polygons delineate the subregions containing the plumes.

The paper has two parts. In the first part, we describe the general principles of lightweight emission quantification and describe the different methods. In the second part, we describe the common framework and interfaces provided by *ddeq* and details of the implementation of the different methods. The application of the *ddeq* library to synthetic CO2M observations for estimating $CO_2$ and $NO_x$ emissions and to TROPOMI observations for estimating $NO_x$ emissions is showcased in several Jupyter Notebooks available in the supplement.

## 2 Theoretical basis

Two families of lightweight approaches exists for emission quantification of hotspots. The first family quantifies emissions from "instantaneous plumes" obtained from single remote sensing images. The second family requires averaging over multiple images taken at different times before quantifying the emissions. The *ddeq* library currently implements four methods for emission estimation, partly in different flavours. The emission quantification methods are (1) Gaussian plume inversion, (2) cross-sectional flux method, (3) integrated mass enhancement method and (4) divergence method. All methods can be applied to single images, although the divergence method typically requires averaging over many images. In the following, the theoretical basis for plume detection, background estimation and for the four quantification methods are summarized briefly. We also briefly cover the conversion of $NO_2$ to $NO_x$ observations and the estimation of annual emissions from individual estimates.



**Table 1.** Symbols used in equations.

| Symbol | Description | Units |
|--------|-------------|-------|
| $A$ | pixel area | $\mathrm{m^2}$ |
| $D$ | decay function | 1 |
| $E$ | emission | $\mathrm{kg\,m^{-2}\,s^{-1}}$ |
| $F$ | gas flux | $\mathrm{kg\,s^{-1}}$ |
| $G$ | Gaussian plume model | $\mathrm{kg\,m^{-2}}$ |
| $H$ | Heaviside step function | 1 |
| $J$ | cost function | - |
| $K$ | eddy diffusion coefficient | $\mathrm{m^2\,s^{-1}}$ |
| $L$ | plume length | m |
| $M$ | integrated mass enhancement | kg |
| $Q$ | emission rate | $\mathrm{kg\,s^{-1}}$ |
| $S$ | sink term | $\mathrm{kg\,m^{-2}\,s^{-1}}$ |
| $V$ | vertical column density | $\mathrm{kg\,m^{-2}}$ |
| $V_{\mathrm{bg}}$ | background vertical column density | $\mathrm{kg\,m^{-2}}$ |
| $b$ | offset of linear background | $\mathrm{kg\,m^{-3}}$ |
| $c$ | decay correction factor | 1 |
| $f$ | $NO_2$ to $NO_x$ conversion factor | 1 |
| $g$ | Gaussian curve | $\mathrm{kg\,m^{-2}}$ |
| $m$ | slope of linear background | $\mathrm{kg\,m^{-3}}$ |
| $p$ | 2D Gaussian surface | $\mathrm{kg\,m^{-2}\,s^{-1}}$ |
| $q$ | line density | $\mathrm{kg\,m^{-1}}$ |
| $r$ | correlation | 1 |
| $t$ | residence time | s |
| $u$ | wind speed | $\mathrm{m\,s^{-1}}$ |
| $x$ | along-plume coordinate | m |
| $y$ | across-plume coordinate | m |
| $z_q$ | detection threshold | 1 |
| $\kappa$ | coefficient in Gaussian plume model | 1 |
| $\mu$ | center shift of Gaussian curve | m |
| $\sigma$ | standard width of Gaussian curve | m |
| $\sigma_V$ | random noise of $V$ | - |
| $\tau$ | decay time | s |
| $\mathcal{P}$ | plume pixels | - |



## 2.1 Plume detection

A critical first step required by most methods is the detection of the plume inside the image, i.e. the detection of the pixels where trace gas columns are enhanced due to emissions from the source of interest. The main purpose is to define a subregion (described by a polygon) that contains the location of the plume to determine the region in the image where the emission quantification method is applied (see Fig. 1). This first step does not only detect the plume location, but also assigns the source location and computes a local coordinate system defining along- and across-plume distances.

Broadly speaking two different approach for plume detection can be defined that can be applied to remote sensing image, both are implemented in *ddeq*. The first approach performs an image segmentation. For example, if thresholding is used, pixels are assigned to the plume if their signal-to-noise ratio (SNR) exceeds a threshold $z_q$

$$\text{SNR} = \frac{V - V_{\text{bg}}}{\sigma_V} \geq z_q \tag{1}$$

where $V$ and $V_{\text{bg}}$ are the total and background vertical column density and $\sigma_V$ is the local noise in the image (Varon et al., 2018; Kuhlmann et al., 2019). The determination of the background, i.e. the vertical column density that would be expected without the presence of the source, is described in the next section. More complex segmentation algorithms apply feature detection using, for example, convolutional neural networks (Finch et al., 2022; Dumont Le Brazidec et al., 2023a). The detected plume can be assigned to one or more sources by checking its overlap with a known list of source locations, for example, from an emission inventory. The boolean mask obtained from the image segmentation can be converted to the polygon shown in Fig. 1 by computing the bounding box in the local coordinate system.

The second approach determines the location of the plume based on the source location and the wind field available from, for example, a meteorological reanalysis product. In the simplest case, the wind vector is taken at the source location and the plume is assumed to be located downstream. It is then possible to draw a rectangular polygon with the along- and across-wind direction. The approach can be extended to simulate the plume location or an ensemble of plume locations with an atmospheric transport model, which can provide a better estimate of the plume location especially further downstream of the source. However, this can get computationally quite expensive and does not qualify as lightweight method anymore.

For each detected plume, a natural coordinate system can be established with along-plume coordinate $x$ and across-plume coordinate $y$. The coordinates can either be computed as distances along and perpendicular to the wind vector. For a curved plume, the coordinates can be computed as arc length along, and distance from, a two-dimensional curve fitted to the detected plume (Figure 1 and Kuhlmann et al., 2020a, for details).

## 2.2 Background estimation

To estimate the emission rate of a source, we are interested in the enhancement above the background

$$V_e(x,y) = V(x,y) - V_{\text{bg}}(x,y). \tag{2}$$

A common approach for estimating the background is applying a low-pass filter (e.g., a median filter) or a normalized convolution after masking enhancements assuming a spatially smooth background field. Alternatively, the background can be estimated





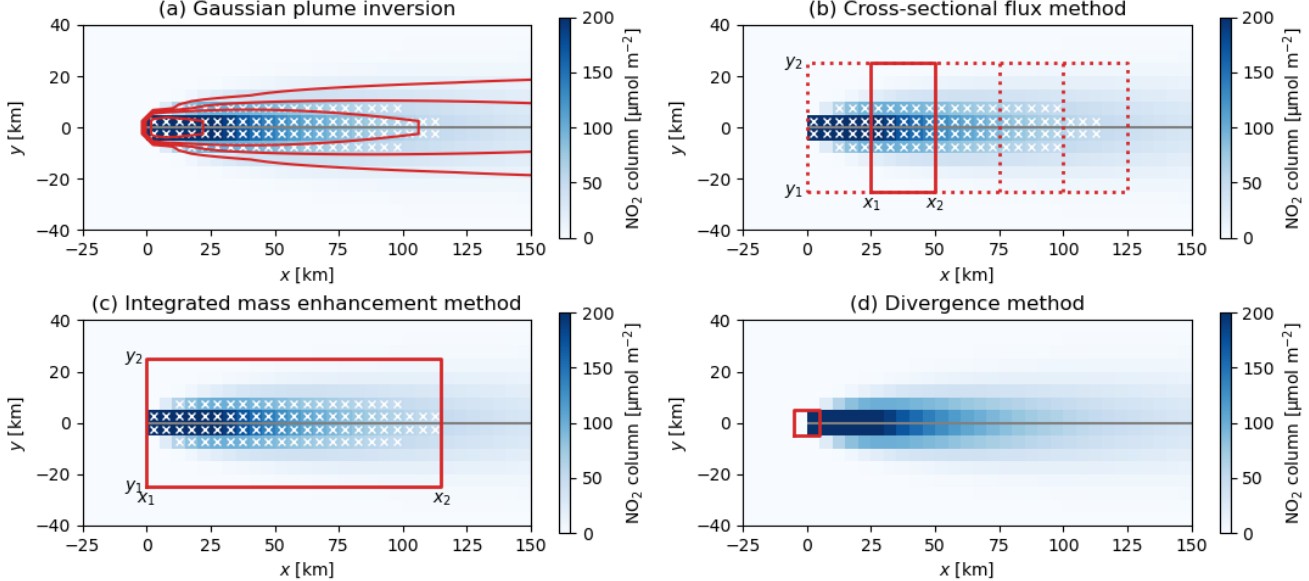

**Figure 2.** Sketch showing the application of the different lightweight methods to an emission plume. Each panel shows an $NO_2$ emission plume simulated by a Gaussian plume model ($Q = 1\,\mathrm{kg\,s^{-1}}$, $u = 5\,\mathrm{m\,s^{-1}}$) including exponential decay ($\tau = 6$ h).

from the pixels upstream of the source. Another approach is explicitly fitting a background term in the emission quantification method, which is possible with the Gaussian plume inversion and the cross-sectional flux method.

## 2.3 Emission quantification methods

Figure 2 illustrates the application of Gaussian plume inversion, cross-sectional flux, integrated mass enhancement and divergence method. In each panel, the pseudo-color map shows the $NO_2$ column densities observed by an imaging spectrometer with 5 km resolution for an emission plume of a source located at the origin. The plume was modelled with a Gaussian plume model with an emission rate of $1\,\mathrm{kg\,s^{-1}}$ ($\approx 32$ kt $NO_2$ $\mathrm{a^{-1}}$), a chemical lifetime of $NO_2$ of 6 hours, and a wind speed of 5 $\mathrm{m\,s^{-1}}$. The white crosses mark pixels where the $NO_2$ column is larger than $50\,\mu\mathrm{mol\,m^{-2}}$. In the following, the theoretical basis

for these methods is described for determining the emissions of a chemical inert gas (e.g., $CO_2$ and $CH_4$) and an exponentially decaying gas (e.g. $NO_2$). The conversion of $NO_2$ to $NO_x$ is discussed in Section 2.4.

### 2.3.1 Gaussian plume inversion

In this method a vertically integrated Gaussian plume model $G$ is fitted to the observed column densities $V$ (e.g., Bovensmann et al., 2010; Nassar et al., 2017). The Gaussian plume model can be written as

$$G(x,y) = \frac{Q\,H(x)}{\sqrt{2\pi}\,u\,\sigma(x)} \exp\left(-\frac{y^2}{2\sigma(x)^2}\right) + V_{\mathrm{bg}}(x,y) \tag{3}$$





with emission rate $Q$, wind speed $u$ and background column $V_{\mathrm{bg}}(x,y)$. $H(x)$ is the Heaviside step function. $x$ and $y$ are the along- and across-plume coordinates. The dispersion in the across-plume direction is modeled by the standard width

$$\sigma(x) = \sqrt{\frac{2Kx^\kappa}{u}} \tag{4}$$

with eddy diffusion coefficient $K$ (in m$^2$ s$^{-1}$). The additional exponent $\kappa$ accounts for possible changes in the dispersion rate along the plume depending on meteorological conditions. This makes it possible to modify the standard expression using $\kappa = 1$ with a power law of the form $\sigma(x) = ax^b$, which has also been used in literature (e.g., Krings et al., 2013). An example of the Gaussian plume is shown by the contour lines in Fig. 2a.

A least squares method can be used to obtain the optimal values for $Q$, $K$, $V_{\mathrm{bg}}$ and $\kappa$ as well as their uncertainties by minimizing the following cost function

$$J(Q, K, V_{\mathrm{bg}}, \kappa) = \|V_{i,j} - G(x_i, y_i)\|_2^2 \tag{5}$$

where $V_{i,j}$ is the observed column density for the pixel with center coordinates $(x_i, y_i)$.

Equation (3) can be used to approximate the emission plumes for species with long lifetimes such $CO_2$, CO and $CH_4$. For species with short lifetimes, the Gaussian plume model needs to be multiplied with a decay term

$$D(x, \tau) = H(x) \exp\left(-\frac{x}{u\tau}\right) \tag{6}$$

where the lifetime $\tau$ is an additional fitting parameter. In the case of $NO_2$, the lifetime is typically about 4 hours.

The Gaussian plume model is valid for a point source, where the source area is smaller than the pixel size. For sources such as cities with dimensions larger than the pixel size, the flux will slowly increase across the source area. An approach to account for this effect is to describe the emissions from the city by an emission map $p(x,y)$ and the change in flux in along-plume direction as the convolution of a map and a decay term

$$G_a(x,y) = G(x,y) \int_{-\infty}^{+\infty} D(x', \tau)\, p(x - x', y)\, dx' \tag{7}$$

The emission map can take the form of uniform surface within city boundaries or a 2D Gaussian surface $p(x,y)$ with the form

$$p(x,y) = \frac{1}{2\pi\sigma_x\sigma_y\sqrt{1-r^2}} \cdot \exp\left(-\frac{(x-x_0)^2}{2\sigma_x^2(1-r^2)} - \frac{(y-y_0)^2}{2\sigma_y^2(1-r^2)} + \frac{r(x-x_0)(y-y_0)}{\sigma_x\sigma_y(1-r^2)}\right) \tag{8}$$

where $(x_0, y_0)$ is the center position, $(\sigma_x, \sigma_y)$ are standard widths and $r$ is the correlation. These parameters can be included in the least square method as additional fitting parameters.

### 2.3.2 Cross sectional flux method

The cross-sectional flux method applies mass conservation by computing the gas flux in the plume ($F$, in kg s$^{-1}$) downwind of the source from wind speed $u$ and line density $q$ (e.g. Varon et al., 2018; Reuter et al., 2019; Kuhlmann et al., 2021), i.e.

$$F = u \cdot q. \tag{9}$$



For a non-decaying gas, the flux is identical to the emission rate $Q$ under the assumption of steady-state conditions and that

turbulent mixing is negligible compared to advective transport in along-plume direction. For a decaying gas, the flux decreases downstream of the source in along plume distance $x$. In this case, the emission rate can be computed by compensating the flux for the along-plume decay:

$$Q = \frac{F(x)}{D(x,\tau)}.$$ (10)

The line density is obtained by integrating the column enhancements in across-plume direction from $y_1$ to $y_2$ at distance $x$:

$$q(x) = \int_{y_1}^{y_2} \left( V(x,y) - V_{\text{bg}} \right) dy$$ (11)

where the interval $[y_1, y_2]$ needs to be sufficiently large to contain the full plume extent.

The line density can simply be obtained by integrating over the enhancements for all pixels within a rectangle (a polygon in case of a curved plume) delimited by $[x_1, x_2]$ and $[y_1, y_2]$ (see Fig. 2b). However, a disadvantage of this approach is that the background needs to be estimated first and subtracted from the observed vertical columns. Another disadvantage is that

missing pixels (e.g. due to clouds) need to be interpolated to obtain the correct line density. An often used alternative approach is therefore fitting a Gaussian curve to all pixels within the rectangle

$$g(y) = \frac{q}{\sqrt{2\pi}\sigma} \exp\left( -\frac{(y-\mu)^2}{2\sigma^2} \right) + my + b$$ (12)

with standard width $\sigma$ and center position $\mu$ to the observations. The background is approximated here by a linear function with slope $m$ and intercept $b$. The Gaussian curve has the advantage that it automatically interpolates missing values. Furthermore,

assuming that different trace gases share the same distribution in lateral direction, the method can be expanded to use the standard width and center position estimated for one trace gas directly when fitting the Gaussian function for another gas. This is particularly attractive for the combination of $NO_2$ and $CO_2$ observations from the future CO2M mission. Since $NO_2$ can be measured with higher precision than $CO_2$ with current remote sensing instruments, images of $NO_2$ can provide a much stronger constraint on the width and position of the plume compared to the much noisier images of $CO_2$ (e.g., Reuter et al.,

2019).

To increase the accuracy of the estimate, fluxes can be computed for multiple polygons downstream of the source. For a non-decaying gas, the estimated fluxes can simply be averaged. For a decaying gas, however, the emission $Q$ can be obtained by additionally fitting the lifetime $\tau$ to the estimated fluxes. For point sources, where the pixel size is larger than the source area, a step function is assumed in the decay term, i.e. the flux increases stepwise from zero to the emission rate $Q$ at the source

location:

$$F_p(x,\tau) = Q \cdot D(x,\tau).$$ (13)

For sources such as cities it is necessary to account for the effect of the source area by describing the emissions from the city, for example, as a Gaussian curve and the change in flux in along-plume direction as the convolution of a Gaussian curve and a



decay term

$$F_a(x, \tau, \mu_a, \sigma_a) = Q \int\limits_{-\infty}^{+\infty} D(x', \tau)\, g(x - x', \mu_a, \sigma_a)\, dx' \tag{14}$$

where $\mu_a$ and $\sigma_a$ are location and standard width of the Gaussian curve describing the extend of the area source. This is identical to the exponential-modified Gaussian method, but applied to a single image (Beirle et al., 2011; de Foy et al., 2015).

### 2.3.3 Integrated mass enhancement

The integrated mass enhancement approach computes the emission rate $Q$ from the integrated total mass enhancement $M$ of the detectable plume $\mathcal{P}_d$ and a residence time $t$ (Frankenberg et al., 2016; Varon et al., 2018). The method can be derived by integrating the Gaussian plume model after subtracting the background (Eq. 3) over a large polygon up to distance $x_2$ (see Fig. 2c):

$$M = \int\limits_{y_1}^{y_2} \int\limits_{x_1}^{x_2} \left( G(x, y) - V_{\text{bg}}(x, y) \right) dx\, dy. \tag{15}$$

If the integration interval in across-plume direction is sufficiently large to contain the full plume, and $0 < x_1 < x_2$, we obtain

$$M = \int\limits_{x_1}^{x_2} \frac{Q}{u}\, dx \tag{16}$$

or

$$Q = \frac{u}{L} M \tag{17}$$

where $u$ is the effective wind speed and $L = x_2 - x_1$ is the length of the detectable plume. Note that the derivation here is different from Varon et al. (2018), who integrated over the detectable plume only and computed $L$ as the length scale defined as the square root of the plume area.

In practise, $M$ can be computed as

$$M = \sum_{(i,j) \in \mathcal{P}_a} \left( V_{i,j} - V_{\text{bg}} \right) \cdot A_{i,j} \tag{18}$$

where $A_{i,j}$ are the pixel areas and the integration area $\mathcal{P}_a$ is obtained by sufficiently expanding the detected plume in across-wind direction to include also pixels with enhancements below the detection limit.

To apply the integrated mass enhancement method to a decaying gas, the decay term needs to be included in the integral, i.e.

$$M = \int\limits_{y_1}^{y_2} \int\limits_{x_1}^{x_2} G(x, y) \cdot D(x)\, dx\, dy. \tag{19}$$





As a result, the emission rate $Q$ can be computed as

$$Q = \frac{1}{c} \frac{u}{L} M \tag{20}$$

where the correction factor $c$ corrects for the gas decay in along-plume direction:

$$c = \frac{u\tau}{L} \left( \exp\left(-\frac{x_1}{u\tau}\right) - \exp\left(-\frac{x_2}{u\tau}\right) \right). \tag{21}$$

### 2.3.4 Divergence method

The divergence method was introduced by Beirle et al. (2011, 2019) for estimating $NO_x$ emissions from TROPOMI $NO_2$ satellite observations. In the CoCO2 project, the method was adapted to estimate $CO_2$ emissions (Hakkarainen et al., 2022). The method is generally applied to a sequence of satellite images rather than a single image.

The divergence method is based on the continuity equation (Jacob, 1999; Koene et al., 2023) at steady state. According to this, the divergence of the flux field $\boldsymbol{F}$ corresponds to the difference between emissions $E$ and sinks $S$:

$$\nabla \cdot \boldsymbol{F} = E - S. \tag{22}$$

The flux $\boldsymbol{F}$ is defined as

$$\boldsymbol{F} = \begin{pmatrix} F_x \\ F_y \end{pmatrix} = \begin{pmatrix} V \cdot u \\ V \cdot v \end{pmatrix} \tag{23}$$

where $V$ is the vertical column density, and $u$ and $v$ are the eastward and northward components of the plume transport speed, respectively, which corresponds to the horizontal wind components weighted by the vertical distribution of the plume concentrations (Koene et al., 2023).

The $NO_x$ sink can be calculated from the $NO_2$ columns as $S = \frac{fV}{\tau}$, where $\tau$ is the $NO_x$ lifetime and $f$ is the constant $NO_x$-to-$NO_2$ ratio. Assumptions about lifetime and $NO_x$-to-$NO_2$ ratio are discussed in the next section.

In case of gases like $CO_2$ with lifetimes much longer than the characteristic timescales of a plume (i.e. much longer than a few hours), the sink term can be neglected. For long-lived gases, however, it is critical to first subtract the atmospheric background before computing the divergence since the flux is not linear with the column $V$ due to the vertical change of wind speed (e.g., Hakkarainen et al., 2016).

To obtain the hotspot emissions $Q$ from the emission map ($E = S + \nabla \cdot \boldsymbol{F}$), a peak-fitting algorithm can be applied that fits a 2D Gaussian surface with a background at each source location ($x_0, y_0$):

$$p(x,y) = \frac{Q}{2\pi\sigma_x\sigma_y\sqrt{1-r^2}} \cdot \exp\left( -\frac{(x-x_0)^2}{2\sigma_x^2(1-r^2)} - \frac{(y-y_0)^2}{2\sigma_y^2(1-r^2)} + \frac{r(x-x_0)(y-y_0)}{\sigma_x\sigma_y(1-r^2)} \right) + p_{\mathrm{BG}} \tag{24}$$

with standard widths ($\sigma_x$, $\sigma_y$), correlation $r$ and a constant background in the divergence flux map $p_{\mathrm{BG}}$.

### 2.4 $NO_2$ to $NO_x$ conversion

Many studies estimate emissions from $NO_2$ observations. However, $NO_2$ is emitted primarily as nitrogen monoxide (NO) and rapidly converted to $NO_2$ in the atmosphere. Emissions are therefore reported as nitrogen oxides ($NO_x = NO_2 + NO$) in $NO_2$



equivalents ($\mathrm{kg\,NO_2\,s^{-1}}$). Since imaging remote sensing instruments only measure $NO_2$ column densities, it is necessary to convert $NO_2$ to $NO_x$ using a $NO_2$ to $NO_x$ ratio $f_V$ representative of vertical columns

$$V_{\mathrm{NO}_x}(x,y) = f_V(x,y) \cdot V_{\mathrm{NO}_2}(x,y) \tag{25}$$

or a ratio $f_Q$ for the estimated emissions

$$Q_{\mathrm{NO}_x} = f_Q \cdot Q_{\mathrm{NO}_2}. \tag{26}$$

If the conversion factor $f_V$ is constant in space, $f_V$ and $f_Q$ are identical. However, the assumption of spatial (and temporal) homogeneity is generally not true for emission plumes and more realistic models are currently being discussed (e.g., Hakkarainen et al., 2023; Meier, 2023).

## 2.5   Estimating annual emissions

Except for the divergence method, the methods described thus far allow us to quantify emissions from a single satellite image. To make statements about emissions over longer periods of time, and to take advantage of the detection of a single source in multiple satellite images, one can compute a temporal average of the various computed emissions. Since the temporal coverage may be sparse and unevenly distributed over the year due to cloud cover and other factors, it may be useful to fill the gaps by making assumptions about the temporal variability. One possibility is to fit a seasonal cycle to the individual estimates using low-order spline to approximate the time-varying emissions and to compute the annual mean emissions by integrating over the cycle (e.g., Kuhlmann et al., 2021). Extrapolating from a few single observations to an annual average is associated with significant uncertainties unless additional information on the true temporal variability is available (Nassar et al., 2022). A further complication is the fact that satellite observations are often performed at the same time of the day providing almost no information on diurnal variability.

## 3   The *ddeq* Python library

In this paper, we describe Version 1.0 of the library, which is provided in the supplement[1]. ddeq is an open source library, whose latest release is available on the Python Package Index (PyPI; https://pypi.org/project/ddeq/). The issue tracker and the development version of the library are available on the project's website: https://gitlab.com/empa503/remote-sensing/ddeq. How to install *ddeq* is described in Appendix A. The documentation of the library is available in the supplement and also published on "Read the Docs" (https://ddeq.readthedocs.io/).

### 3.1   General framework

The library consists of four main components as shown in Figure 3. (1) The data input component provides functions for reading remote sensing images (e.g., S5P/TROPOMI observations and synthetic CO2M observations), hotspot locations (e.g., CoCO2

---

[1]Note that in the preprint version, the *ddeq* version is denoted as release candidate (Version 1.0-rc1).




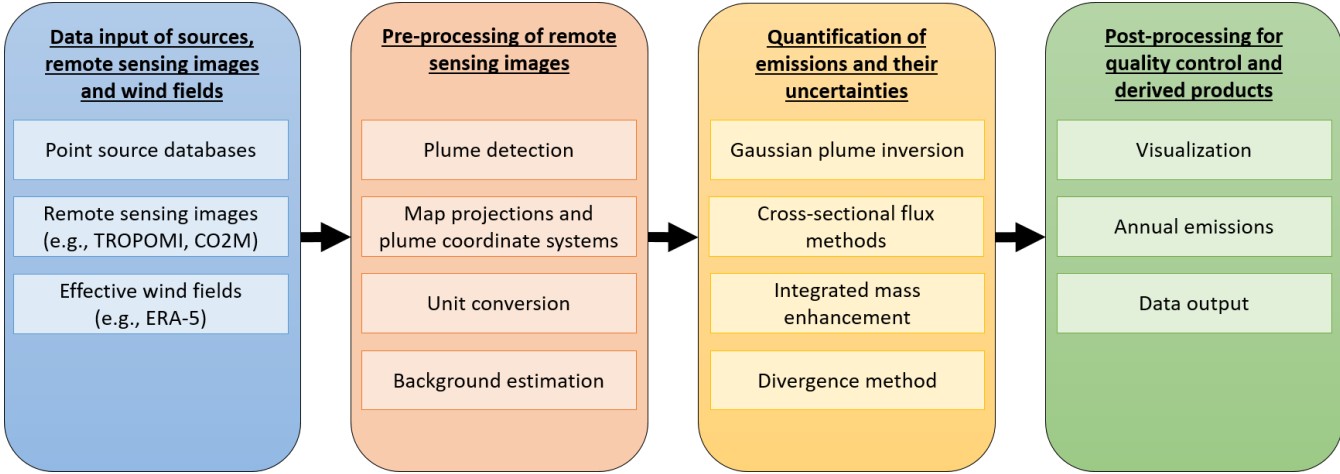

**Figure 3.** Data flow diagram illustrating the interactions between the different components in *ddeq*.

point source database, Guevara et al. (2023)) and (effective) wind fields (e.g., ERA-5 reanalysis product). (2) The second component provides pre-processing of the data, which includes plume detection, conversion between coordinate systems, unit conversions, and estimation of the background field. (3) To quantify the emissions, five modules are provided implementing a Gaussian plume inversion (hereafter abbreviated as GP, in `ddeq.gauss`), two cross-sectional flux methods (hereafter CSF, in `ddeq.csf` and LCSF in `ddeq.lcsf`), integrated mass enhancement (hereafter IME, in `ddeq.ime`) and divergence method (hereafter DIV, in `ddeq.div`). (4) Finally, the library provides functions for post-processing, which includes methods for estimating annual emissions from individual estimates, for converting $NO_2$ to $NO_x$ emissions, for visualizing the results and for writing data output in standardized NetCDF format.

Figure 4 shows a simplified code example demonstrating the individual steps required for estimating the $CO_2$ and $NO_x$ emissions of the Jänschwalde power plant in Germany. The Jupyter Notebook with the full example is part of the *ddeq* library ('notebooks/tutorial-introduction-to-ddeq.ipynb'). First, the data input step corresponds to reading the location of sources, synthetic CO2M data and ERA-5 wind fields. ERA-5 file were downloaded and prepared by the *ddeq* library (cf. Section 3.2). Second, a plume detection algorithm is used to locate the plume in the satellite image. A center curve is fitted to the data and natural coordinates are computed for the detected plume of each source. Third, the data are prepared for emission quantification by estimating and subtracting the background field and converting the $CO_2$ and $NO_2$ columns to $\mathrm{kg\,m^{-2}}$. Fourth, $CO_2$ and $NO_x$ emissions are estimated using the cross-sectional flux method as an example. The estimated $NO_2$ emissions are converted to $NO_x$ using a conversion factor of 1.32 (in $\mathrm{kg\,NO_2\,s^{-1}}$). Finally, the results are saved as a NetCDF file and the data are visualized (see Fig. 5). In the following, the different components are described in more details. The implementation details are available in the documentation and the code itself in the supplement.





```
# (1) read hot spot locations, remote sensing data, winds
sources = ddeq.misc.read_point_sources()
rs_data = ddeq.smartcarb.read_level2('Sentinel_7_CO2_2015042311_o1670_l0483.nc',
                                     co2_noise_level='low', no2_noise_level='high')
winds = ddeq.wind.read_wind_at_sources(rs_data.time, sources, product='ERA-5')

# (2a) plume detection from NO2 observations
rs_data = ddeq.dplume.detect_plumes(rs_data, sources, variable='NO2', variable_std='NO2_std',
                                    filter_type='gaussian', filter_size='0.5')

# (2b) creation of plume-following coordinates
rs_data, curves = ddeq.plume_coords.compute_plume_line_and_coords(rs_data, radius=25e3)

# (3) background estimation and unit conversion
rs_data = ddeq.emissions.prepare_data(rs_data, 'CO2')
rs_data = ddeq.emissions.prepare_data(rs_data, 'NO2')

# (4) emission quantification (here: cross-sectional flux method)
results = ddeq.csf.estimate_emissions(rs_data, winds, sources, curves, method='gauss',
                                      gases=['CO2', 'NO2'], f_model=1.32)

# (5) data output and visualization
results.to_netcdf(output_filename)
ddeq.vis.plot_csf_result(['CO2', 'NO2'], rs_data, winds, result, curves, 'Janschwalde')
```

**Figure 4.** Example of *ddeq* code for applying the CSF method to estimate $CO_2$ and $NO_x$ emissions of the Jänschwalde power plants in Germany. The full code is available in the library as a Jupyter Notebook ('notebooks/tutorial-introduction-to-ddeq.ipynb').

## 3.2 Data input

*ddeq* requires that the location of sources used known prior to estimating the emissions. The location and type of sources is therefore an important input for plume detection and emission quantification. *ddeq* makes extensive use of the *xarray* package (Hoyer and Hamman, 2017) for data handling to combine arrays with attributes. Sources are read from a CSV file into an 'xarray.Dataset', which contains the source names ('source'), longitudes ('lon_o'), latitudes ('lat_o'), labels for visualization ('label') and source types ('type', which is currently either 'city' or 'power plant'). *ddeq* maintains a small list of

sources as a comma-separated values (CSV) file that primarily contains cities and power plants used in previous studies by the developers. User-defined files containing other sources can be prepared in the same format. The file can be read with the 'ddeq.misc.read_point_sources' function. In addition, *ddeq* can read the comprehensive CoCO2 global emission point source database (Guevara et al., 2023) using 'ddeq.coco2.read_ps_catalogue'. The catalogue is provided together with the library.



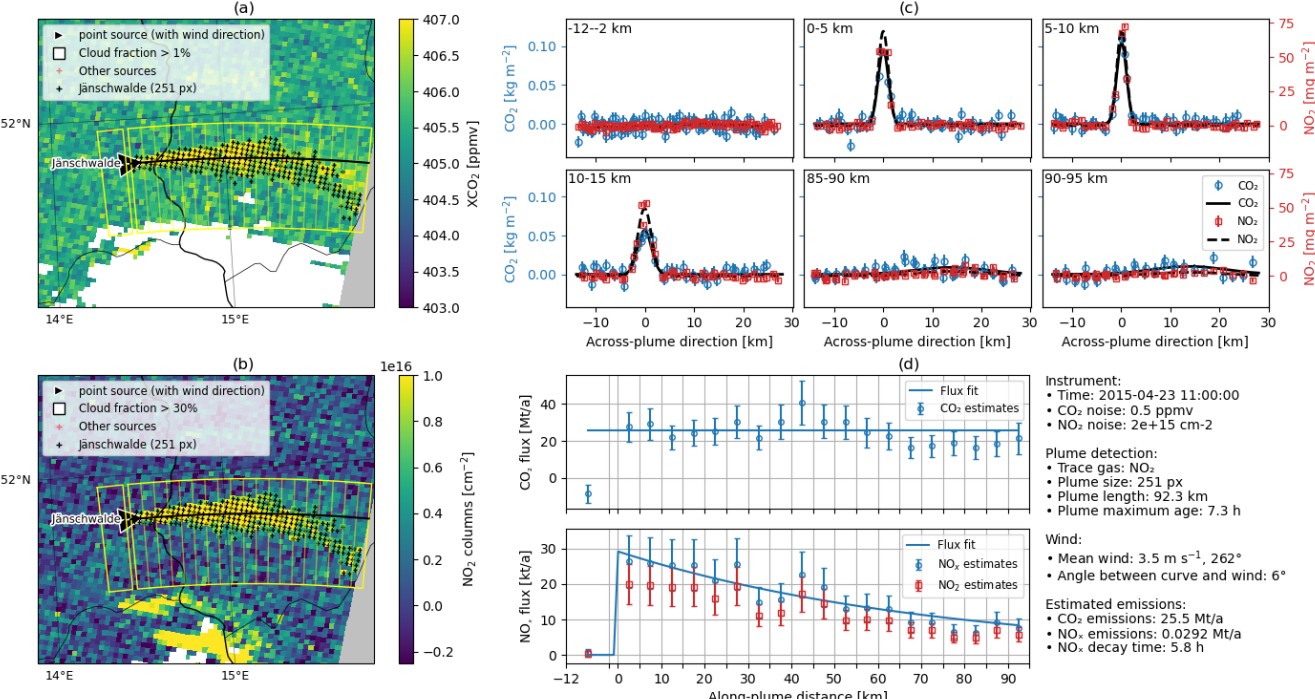

**Figure 5.** Plot created by the 'ddeq.vis.plot_csf_result' command in the short example. Panel (a) and (b) show the $CO_2$ and $NO_2$ plume from the Jänschwalde power plant in the synthetic CO2M images. Panel (c) shows the $CO_2$ and $NO_2$ columns in across-plume direction for different along-plume distances with the two fitted Gaussian curves for computing the line densities. Panel (d) shows $CO_2$ and $NO_x$ flux in along-plume distance. The estimated $CO_2$ and $NO_x$ emissions were $28.5\,\mathrm{Mt\,a^{-1}}$ and $37.4\,\mathrm{kt\,a^{-1}}$ for this example.

Remote sensing images are provided by airborne and space-based imaging spectrometers. *ddeq* handles images as 'xar-
ray.Dataset' with variables providing the trace gas columns and their uncertainties (e.g. 'CO2' and 'CO2_precision') that need
to have a 'units' attribute for automatic unit conversion and a 'noise_level' attribute that is used as random uncertainty by
the plume detection algorithm. In addition, the central longitude and latitude of the pixels need to be provided as 'lon' and
'lat'. If trace gases are provided as column-averaged mole fractions, surface pressure needs to be provided as 'psurf' for unit
conversion. Units are converted using the *ucat* Python library (https://pypi.org/project/ucat/).

*ddeq* provides functions for automatically downloading and cropping TROPOMI $NO_2$ data for a given list of sources
('ddeq.dowload_S5P'). Furthermore, the library can read the synthetic CO2M and Sentinel-5 data from the SMARTCARB
dataset (Kuhlmann et al., 2020b) as well as the simulations from the library of plumes generated in the CoCO2 project (Koene
and Brunner, 2022). The synthetic datasets with known true emissions were used in the CoCO2 project for method development
and benchmarking (Santaren et al., 2023).

The final important input for emission quantification are wind fields, from which a representative transport speed of the
trace gas within the plume is computed. *ddeq* provides functions for reading and downloading ERA-5 reanalysis fields as well



as reading wind fields from the SMARTCARB project ('ddeq.era5' and 'ddeq.wind'). Wind speed and direction are either provided at the location of the source ('ddeq.wind.read_at_sources') or as two-dimensional field that can also be spatially interpolated to the remote sensing image pixels, for example, for computing fluxes on Level-2 data ('ddeq.read_field'). Transport speeds can be computed by averaging the winds over a range of pressure levels, over the depth of the planet boundary layer, or as weighted averages weighted by a vertical profile such as a typical emission profile of power plants (Brunner et al., 2019). Users can also use their own wind data from other data sources.

### 3.3 Pre-processing

Data pre-processing includes a plume detection algorithm, conversion between coordinate systems, unit conversions and estimation of the background field. Which pre-processing steps are required or optional depends on the individual method.

One main pre-processing step is an algorithm for identifying the (a priori) location of the emission plume in the remote sensing image. *ddeq* implements an image segmentation algorithm for plume detection that is used as a pre-processing step by the GP, CSF and IME method. As an alternative, the LCSF method determines the (a priori) plume location from the source location and the wind vector, which is currently part of the method's implementation. The divergence method does not require information about plume locations.

The image segmentation algorithm is described in detail in Kuhlmann et al. (2019, 2021). In short, the algorithm generates a boolean mask which is true where column densities are significantly enhanced above the background using Eq. (1). The signal is computed as the difference between the local mean and the background field. The noise $\sigma_V$ is computed from the random and systematic uncertainty of the vertical column densities and uncertainties in the background. The local mean is computed by applying a uniform or a Gaussian filter to the image. The background field is computed using a median filter with a kernel that is large enough to contain areas outside the plume. The threshold $z_q$ is computed from the probability $q$ that the local mean is larger than the background given the uncertainty $\sigma_V$ based on a statistical z-test. In the boolean mask, neighboring pixels are connected to regions and regions overlapping with known sources are labeled as plumes. The 'ddeq.dplume.detected_plumes' function is used for applying the algorithm to the remote sensing images.

To create the natural coordinate system and compute the plume length, a center curve can be fitted to the plume mask using the 'ddeq.plume_coords.compute_plume_line_and_coords' function. The center curve is described by two parabolic polynomials from which the along-plume coordinate $x$ is computed as arc length from the source and the across-plume coordinate $y$ is computed as the distance from the center curve (see Kuhlmann et al., 2020a, for details). The plume length is the arc length from the source to the most distant detectable pixel. Prior to fitting the center curve, longitude and latitude are converted to eastings and northings using a Cartesian coordinate reference system object from the 'cartopy' library (Met Office, 2010 - 2015). If pixel corners are provided by the input dataset, the function also computes the pixel size (in m$^2$), which is required, for example, by the integrated mass enhancement method.

Finally, 'ddeq.emissions.prepare_data' can be used to estimate the background field and convert all trace gas fields to mass columns (in $\mathrm{kg\,m^{-2}}$) using the *ucat* Python library (Kuhlmann, 2022). *ddeq* implements a function for estimating the background field from pixels surrounding a detected plume ('ddeq.background.estimate'). The function masks all pixels where





the signal-to-noise ratio is larger than the threshold $z_q$ (using the results from the plume detection algorithm) and applies a normalized convolution to estimate the background. Alternatively, the background can also be fitted directly by some emission quantification method.

## 3.4 Emission quantification

### 3.4.1 Overview

Methods that are applied to single images all use the same order of parameters to estimate the emissions of 'sources':

```
results = ddeq.{method}.estimate_emissions(
    rs_data, winds, sources,
    [curves, gases, priors],
    variable='{gas}_minus_estimated_background_mass',
    [...]
)
```

where '{method}' can be 'gauss', 'lcsf', 'csf', 'ime' for Gaussian plume inversion, (light) cross-sectional flux method and integrated mass enhancement method, respectively. Each method iterates over all sources provided by the 'sources' dataset and estimates an emission if the source is inside the image. The method is applied to the variable in the remote sensing dataset ('rs_data') given by the 'variable' parameter (currently not implemented for LCSF and DIV). The default string is '{gas}_minus_estimated_background_mass', which is the default variable name created by the pre-processing after subtracting the estimated background and converting to mass columns with units of $\mathrm{kg\,m^{-2}}$. It is possible to provide a list of up to two gases for the CSF, GP and LCSF method (e.g, 'gases = ['NO2', 'CO2']'). In this case, either both gases are fitted simultaneously (CSF) or after each other where the results from the first fit are used to constrain the second fit (GP and LCSF). The IME method can only be applied to a single gas. The 'gas' placeholder in the 'variable' parameter will be replaced with the names in 'gases'.

The divergence method works on a series of remote sensing images and wind fields, which is called as

```
results = ddeq.div.estimate_emissions(
    datasets,
    wind_folder,
    sources, ...
)
```

where datasets is a class with a 'read_date' method that returns a list of satellite datasets for a given date for which the divergence field is computed and 'wind_folder' is the path to a folder containing, for example, ERA-5 or COSMO wind fields. Examples for the dataset are the 'Level2Dataset' class in the 'smartcarb' module or the 'Level2TropomiDataset' in the 'sats' module.




All methods return a results dataset, which is a 'xarray.Dataset' with at least the variables '{gas}_estimated_emissions'
and '{gas}_estimated_emissions_precision' with dimension 'source' that can be saved using the 'to_netcdf' method. The
365 CSF method stores the results datasets using the 'ddeq.misc.Result' class, which inherits from 'dict' to handle dimensions of
different size between sources. It has the methods 'to_netcdf' and 'from_file' to write and read the results. The results will
be saved as NetCDF files using a group for each source. The implementation of the 'ddeq.misc.Result' class is necessary as
NetCDF groups are currently not supported by xarray.

### 3.4.2 Gaussian plume inversion (GP)

The Gaussian plume inversion method implemented in *ddeq* fits Eq. (3) to the detected plume. The fit parameters are $Q$, $V_{\mathrm{bg}}$, $K$,
$\kappa$ and coefficients of the center line. The wind speed is taken from the wind dataset, which, when using 'ddeq.wind.read_at_sources',
is the effective wind speed at the source.

The plume center curve is described by a second order Bezier curve which has three control points (one centred at the
known plume source location, the other two are fitted parameters for the Gaussian curve), initialized along the curve as already
obtained in pre-processing. The reason for using a Bezier curve is that it behaves smoothly with respect to small changes in the
control points, which is required for stabilizing the least-squares fit. In case of a decaying gas, it is possible to fit a decay time
$\tau$ by setting the 'fit_decay_times' to true.

The inversion consists of three simple Levenberg-Marquardt least-squares fitting steps. In the first fit, only the center curve,
$Q$ and $\kappa$ are optimized. In the second fit, $K$ and $\tau$ are optimized and in the third fit, $Q$ is optimized again. The implementation
of three fits decreases the computation time and avoids overfitting.

The initial (prior) parameters for $Q$ and $\tau$ need to be provided as a dictionary for each source and trace gas:

```
priors = {
    source: {
      'CO2': {'Q': 1000.0, 'tau': 1e10},
'NO2': {'Q': 1.0, 'tau': 4.0*3600},
      ...
}
```

where `Q` is the source strength (in $\mathrm{kg\,s^{-1}}$) and `tau` is the decay time in seconds. Other parameters are set to typical values
in the 'ddeq.gauss.generate_params' function. If two gases are provided, the values fitted for the first gas are used as initial
conditions for the second gas (except for $Q$, which is reinitialized to the prior emission rate for that source). To constraint the
fit for the second gas, we only allow a small amount of deviation around the previously obtained Gaussian plume parameters
(see 'ddeq.gauss.gaussian_plume_estimates' for details).

The precision (aka: random uncertainty) of the Gaussian plume estimate is computed as

$$\sigma_Q = \sqrt{\sigma_{Q,\mathrm{fit}}^2 + \sigma_u^2 \left( \frac{Q_{\mathrm{fit}}}{u} \right)^2} \tag{27}$$





where $\sigma_{Q,\text{fit}}$ is the estimated standard deviation of the fitted emission data and the second term accounts for the uncertainty of the wind speed as provided by the winds dataset ('wind_speed_precision').

Estimates are rejected when no fit is found, no standard deviation is estimated (i.e., if no good fit is possible), or when the emission rate is smaller than 0.1 times or larger than 1.9 times the prior expected emission rate (i.e. using Q±90% uncertainty). These filters is currently fixed in the code. More flexible filters will be implemented in future that are common to all methods.

### 3.4.3 Cross sectional flux methods (CSF)

The CSF method is one of two different cross-sectional flux methods implemented in the library. The CSF method was originally developed in the SMARTCARB study (Kuhlmann et al., 2020a, 2021). It computes line densities in multiple polygons downstream of the source (see polygons in Fig. 5). Depending on the 'model' parameter, line densities are either obtained by fitting a Gaussian curve (Eq. 12) using 'model="gauss"', or integrating pixels inside the polygon using 'model="sub_areas"'.

The latter method divides the sub-polygons in across-plume direction, computes the mean for each sub-polygon, and then integrates the mean values for each sub-polygon. This is done to account for missing values in the image. For $NO_2$ observations, the 'f_model' parameter can be provided to convert $NO_2$ to $NO_x$ line densities.

The line densities are converted to fluxes using the provided wind speeds at the sources. Emissions are obtained by fitting Eq. (13) or Eq. (14) to the fluxes. For a non-decaying gas, the decay function is replaced by a Heaviside step function, which

is 0 upstream and 1 downstream of the source. The uncertainty of the cross-sectional flux method is computed by propagation of uncertainty from the single sounding precision that determines the uncertainty of each line density, which determines the uncertainty of the fitting parameters ($Q$ and $\tau$). Note that we assume that the wind speed uncertainty is independent of the number of pixels and the length of the plume.

To remove problematic cases, estimates are excluded if the angle between wind speed and center curve is larger than 45

degrees, which often indicates erroneous plume detection. Estimates are also rejected if more than 5 pixels are detected upwind of the source.

### 3.4.4 Light cross sectional flux method (LCSF)

The LCSF method is derived from the method originally developed by Zheng et al. (2020) to estimate the $CO_2$ emissions of Chinese cities and industrial areas from OCO-2 data. The method has then been adapted for routine and automatic estimation of

420 isolated clusters of $CO_2$ emissions worldwide (Chevallier et al., 2020) and used to study the temporal variability of emissions using several years of OCO-2 and OCO-3 data (Chevallier et al., 2022)

Similar to the CSF method, the LCSF method computes line densities by interpolating the pixels contained within polygons downstream of the source by a Gaussian function. The LCSF method uses the wind vector to construct a polygon that is 100 km wide in across-wind (perpendicular) direction and which extends downwind the source over a distance equal to the distance

travelled by the wind in one hour. A two-dimensional wind field read with 'ddeq.wind.read_field' is used to determine the wind vector and for computing the wind speed used later for computing the flux. The 'fit_backgrounds' parameter can be used to determine if a linearly changing background should be added to the Gaussian curve. Additional parameters are passed using a





dictionary ('lcs_params'). For example, a $NO_2$ to $NO_x$ conversion factor can be defined using the ''f_NOx_NO2' key (default: 3.5).

The uncertainty in the estimates provided by the LCSF method is computed by propagation of the uncertainty of the amplitude of the fitted Gaussian function. Several quality checks remove potentially unrealistic estimates: the fitting window should contain enough data pixels (default: 50 pixels) and the selected enhancements should have sufficient amplitude (i.e. larger than the standard deviation of the values in the polygon), the uncertainty ($1\sigma$) of the fitted width of the Gaussian function should be larger than 1 km and smaller than 5 km, and the estimated emissions should be larger than 'min_est_emis' (default: 0.0) and

smaller than 'max_est_emis' (default: infinity), which when calling 'ddeq.lcsf.estimate_emissions' are either provided as input parameters or as prior values similar to the implementation for the Gaussian plume inversion (i.e. using $0.1\,Q$ and $1.9\,Q$).

### 3.4.5    Integrated mass enhancement (IME)

To first identify the location of the plume, the IME method uses the plume detection algorithm with the same settings as for the CSF and GP method. IME also requires the computation of the background field, the center line and the along- and across-

plume distance for each pixel. In the across-plume direction, the integration interval is obtained by either computing the convex hull of the detected plume or by applying a binary dilation to the boolean mask of detected pixels. Both computations are carried out as part of the pre-processing inside the 'ddeq.plume_coords.compute_plume_line_and_coords' function. In along-plume direction, the integration interval is defined by the 'L_min' and 'L_max' parameters. The default values are $L_{\min} = 0$ (i.e. the source location) and $L_{\max}$ being set to the arc length of the most distant pixel in the integration area minus 10 km.

Missing values are interpolated using normalized convolution. The plume is discarded when more than 25% of the pixels in the detected plume have been obtained through this gap-filling. The effective wind speed is taken from the provided wind speeds at the source location. A decay time can be provided to compute the decay time correction term (Eq. 21).

The uncertainty of the emissions is calculated by propagation of uncertainty from the random uncertainty of the gas columns and the wind speed.

### 450    3.4.6    Divergence method (DIV)

In *ddeq*, the divergence method is applied to each source individually. The remote sensing images are provided as a dataset class as described above where images are read for each date between 'start_date' and 'end_date'. To access the wind fields, the folder containing the wind files is provided and a filename pattern (e.g., 'ERA5-gnfra-%Y%m%dt%H00.nc').

The divergence method is performed in two steps. In the first step, the average divergence $\nabla \cdot F$ and sink $S$ fields are

calculated from the vertical column densities and wind fields. In the second step, the peak fitting is applied to derive the hotspot emissions.

In the first step, the $XCO_2$ is first denoised using a median filter and depending on the choice of the user, either the local or the regional background is removed. Each pixel is associated with interpolated $u$ and $v$ wind components from the provided wind fields and the vector field $\boldsymbol{F}$ is derived using second order accurate central differences as implemented in 'numpy.gradient'.

All the data is gridded to a regular km-grid defined by the user by the input parameters 'lon_km', 'lat_km' and 'grid_resol'



given in km. The divergence of the vector field is calculated before averaging over all available images. The computation of the divergence before averaging is preferred for remote sensing images with data gaps, for example, due to clouds (Hakkarainen et al., 2022; Koene et al., 2023). For $NO_x$, the sink term is calculated from the averaged vertical column densities assuming a lifetime of 4 h. For $CO_2$, the sink term is not considered.

In the second step, the peak fitting is performed by performing a least squares fit between the averaged emission field $E = S + \nabla \cdot \boldsymbol{F}$ and the peak fitting function Eq. (24). The optimization is first done using the Nelder–Mead method from the *scipy* library. *ddeq* also implements the adaptive Metropolis algorithm (Haario et al., 2001) for sampling the posterior distribution (assuming non-informative prior). In addition to the peak fitting parameters, the statistical error of the emission field is also considered.

## 3.5  Post-processing

The post processing step provides functions for data visualization ('ddeq.vis') and for estimating annual emissions from individual estimates ('ddeq.timeseries'). In addition, it is possible to apply scaling factors to estimated emissions to convert $NO_2$ to $NO_x$ emissions.

To convert $NO_2$ to $NO_x$ emissions, it is possible to multiply all estimated $NO_2$ emissions in the 'results' dataset, i.e. variables that start with "NO2" and have units of "kg s-1", to $NO_x$ emissions using the scalar $f$. The function 'convert_NO2_to_NOx_emissions(results, f=1.32)' is implemented in the 'ddeq.emissions' module. A more complex $NO_2$ to $NO_x$ conversion will be developed in future to account for non-constant conversion factors.

*ddeq* provides functions for fitting a seasonal cycle using a cubic C-spline with periodic boundary conditions to a time series of estimated emissions ('ddeq.timeseries'). Annual emissions and their uncertainties are obtained by integrating the seasonal
cycle as shown in the following code:

```
fit, model, _, _, _ = ddeq.timeseries.fit(
    times,
    estimates,
    estimates_precision
)
annual, annual_precision = model.integrate(
    fit['x'],
    fit['x_std']
)
```

A detailed example is available as Jupyter Notebook ('example_annual_emissions.ipynb').

Finally, *ddeq* provides functions for visualizing the remote sensing images, the plume detection, and the results of the emission quantification using 'ddeq.vis.plot_{method}_result'. The library includes several Jupyter Notebooks with demonstrations for the different methods.



## 4 Conclusions

There have been many studies quantifying the emissions of hotspots in remote sensing images in the past and we expect significantly more studies in future with the new generation of imaging satellites. The ability to monitor emissions of air pollutants and greenhouse gases is an important capability of remote sensing instruments. As information on emissions from hotspots can be sensitive, it is essential that the emission estimates are reliable, i.e. that the methodology used can be verified, for which an open-source library will be an invaluable tool.

*ddeq* is a community library that is open to new users and developers. The Jupyter notebooks with tutorials and examples make it easy for new users to learn how to apply the library to different datasets. *ddeq* is hosted as an open-source project on GitLab making it possible for users and developers to submit bug reports and feature request. Finally, developers can further expand existing methods in *ddeq* or implement new methods. *ddeq* makes it possible to compare these additions with existing methods in a reproducible way, for example by using the same input data, increasing the reliability of estimates of

anthropogenic emissions of air pollutants and greenhouse gases.

The *ddeq* library provides a general framework where new methods and options can easily be implemented. It will be further developed in the future, for example, in the CORSO project, where *ddeq* will be used for quantifying CO and $NO_x$ emissions from Sentinel-5P/TROPOMI and other satellite instruments. Furthermore, *ddeq* will be used for quantifying $CH_4$ emissions from the new airborne ARES imaging spectrometer implemented by a consortium of Swiss research institutes led by the

510 University of Zurich. Future updates will further harmonize the different implementations allowing for more flexibility and even better comparability between the different methods. It is also planned to provide support for machine-learning models such as the plume segmentation algorithm developed by Dumont Le Brazidec et al. (2023a).

One main goal of the CoCO2 project is the development of prototype systems for anthropogenic emission monitoring system. The *ddeq* Python library for data-driven emission quantification presents such a prototype for emission monitoring of hotspots

using lightweight approaches. It was developed and used in the CoCO2 project to benchmark different emission quantification methods with the aim to identify the most suitable and accurate approaches to be implemented in the prototype of the European $CO_2$ monitoring system (Hakkarainen et al., 2023; Santaren et al., 2023).

*Code and data availability.* The *ddeq* version 1.0 described in this document is available in the supplement. The code repository is available on Gitlab.com (https://gitlab.com/empa503/remote-sensing/ddeq). The SMARTCARB dataset is available on Zenodo (https://doi.org/10.

5281/zenodo.4048227).

## Appendix A: Installation and interactive computing environment

The latest version of the *ddeq* library can be installed using Python's package manager pip:

```
python -m pip install ddeq
```



The development version of *ddeq* can be cloned using

```
git clone https://gitlab.com/empa503/remote-sensing/ddeq.git
```

and then installed using pip. For the development version it is useful to install it editable (-e option):

```
pip install -e ddeq/
```

The version described in this paper is available in the supplement. The version can be installed using 'conda' and 'pip' with following steps:

```
# create Python environment
conda create -n ddeq-test python=3.9
conda activate ddeq-test

# install additional packages
conda install jupyterlab
conda install pycurl

# unzip tar.gz file in the supplement and install using pip
tar -xzvf ddeq-1.0rc1.tar.gz
python -m pip install ddeq-1.0rc1/

# start JupyterLab
jupyter lab --notebook-dir .
```

The tutorial and examples are included in the 'notebooks' folder.

*Author contributions.* The paper was written by GK with input from all the co-authors. GK developed the original *ddeq* library in the SMARTCARB project and merged all implementations from the CoCO2 and CORSO project; EK developed the Gaussian plume inversion method; SM implemented the code for downloading and pre-processing the ERA5 and S5P/TROPOMI data; DS, GB and FC developed the light cross-sectional flux method JH, JN and LA implemented the divergence method. The project was coordinated by DB, GB, GK and JT.

*Competing interests.* The authors declare no competing interests.

*Acknowledgements. ddeq* was developed in the context of the ESA-funded SMARTCARB project (no. 4000119599/16/ NL/FF/mg), the H2020 CHE and CoCO2 projects (no. 776186 and no. 958927) and the Horizon Europe CORSO project (no. 101082194) with additional





funding by the Swiss State Secretary for Education, Research and Innovation (SERI). We like to thank the ICOS Carbon Portal for providing access to their JupyerLab servers, which were used for code development and data sharing.



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
