# Peer review of "The *ddeq* Python library for point source quantification from remote sensing images (Version 1.0)"

_EGUsphere, 2023_

## Referee Comment (RC1)

General comments:

The manuscript entitled, "*The ddeq Python library for point source quantification from remote sensing images (Version 1.0)*," summarizes the capabilities of a new open-source Python library designed to estimate greenhouse gas emissions rates from point sources and cities using a variety of different methods. This software represents a substantial contribution to the field and the lightweight methods used are valid and described by an appropriate amount of detail. The library was relatively easy to install, and the example Jupyter Notebooks are useful. The manuscript is very well-written, and I recommend publication after minor comments are addressed.

1. The manuscript would benefit from more discussion on wind, as wind speed and direction errors are one of the largest contributors to emission rate errors. The code can download ERA5 winds, but it is not clear where wind_speed_precision comes from (presumably ERA5 as well) or if it is a good estimate of the true error. Additionally, a brief discussion on the effective wind speed (*u)* would be appreciated. For example, how it is calculated and how it varies depending on instrument pixel resolution.

2. Some discussion on background uncertainty, where appropriate, would also improve the manuscript. S3.3 mentions "uncertainties in the background" but more detail is needed, as this is another important source of error in emission rate estimates. Background uncertainty does not appear to be included in the uncertainty terms at this time, e.g., in S3.4.5, "The uncertainty of the emissions is calculated by propagation of uncertainty from the random uncertainty of the gas columns and the wind speed."

3. This is certainly beyond the scope of the manuscript and Version 1.0 of *ddeq*, but it would be great to have support for OCO-2/3 $XCO_2$ data in a future release!

Specific comments:

**S2.3.2: "Furthermore, assuming that different trace gases share the same distribution in lateral direction, the method can be expanded to use the standard width and center position estimated for one trace gas directly when fitting the Gaussian function for another gas. This is particularly attractive for the combination of NO2 and CO2 observations..."**

Can the same width always be used when combining $NO_2$ and $CO_2$, or do you need to take into account the $NO_2$ decay as you get further from the source?

**S2.5: "Nassar et al., 2022"**

You may also consider citing Hill and Nassar, 2019 (https://www.mdpi.com/2072-4292/11/13/1608)

**S3.4.2: "or when the emission rate is smaller than 0.1 times or larger than 1.9 times the prior expected emission rate"**

What about a scenario where the prior expected emission rate is highly uncertain (e.g., in an undeveloped nation)? Perhaps these limits would be too constraining.

**S3.4.6 L457: "In the first step, the XCO₂"**

Is this just for $CO_2$, or also $NO_2$? It's also the only time $XCO_2$ is mentioned in the manuscript, so you'd need to define it.

**S3.4.6: "the statistical error of the emission field is also considered."**

Perhaps explain this a bit more.

**S4: "As information on emissions from hotspots can be sensitive"**

Please be more specific on what you mean by sensitive here.

**tutorial-introduction-to-ddeq.ipynb:**

Running the latest release of *ddeq* (via $pip install ddeq), cell 16 fails with "AttributeError: Module 'scipy' has no attribute 'vectorize'". It looks like era5.py L500 should have "np.vectorize" instead of "scipy.vectorize". This causes issues in other notebooks too.

Technical corrections:

L94 "distance along **or** perpendicular"
L108 "**chemically** inert"
L176 "are **the** location"
L176 "the **extent** of the area source"
L259: define "LCSF"
L275: "location of sources used **is** known"
L277: define "CSV" here and not on L280
Figure 5: the $CO_2$ and $NO_x$ emission estimates in the figure do not match those listed in the figure caption.
L308: "GP, CSF and IME method**s**"
L333: "method**s**"
L360: define "COSMO"
L390: "to **constrain**"
L394: " random uncertainty"
L477: "in **the** future"
L496: "in **the** future"
L507: define "CORSO"
L509: define "ARES"
L513: "prototype systems for anthropogenic emission monitoring "

---

## Author Comment (AC1)

**Reply to reviewers' comments**

Kuhlmann et al.: The ddeq Python library for point source quantification from remote sensing images (Version 1.0)

*Author comments in italic blue.*

*Manuscript changes in italic red.*

**RC1: Robert Roland Nelson**

General comments:

The manuscript entitled, "The ddeq Python library for point source quantification from remote sensing images (Version 1.0)," summarizes the capabilities of a new open-source Python library designed to estimate greenhouse gas emissions rates from point sources and cities using a variety of different methods. This software represents a substantial contribution to the field and the lightweight methods used are valid and described by an appropriate amount of detail. The library was relatively easy to install, and the example Jupyter Notebooks are useful. The manuscript is very well-written, and I recommend publication after minor comments are addressed.

*We like to thank Robert Nelson for their positive and constructive comments. In the following, we address his comments point by point.*

1. The manuscript would benefit from more discussion on wind, as wind speed and direction errors are one of the largest contributors to emission rate errors. The code can download ERA5 winds, but it is not clear where wind_speed_precision comes from (presumably ERA5 as well) or if it is a good estimate of the true error. Additionally, a brief discussion on the effective wind speed (u) would be appreciated. For example, how it is calculated and how it varies depending on instrument pixel resolution.

   *We have added a new section on the effective wind in the theory section. In our current version, wind precision is hard-coded as 1 m/s, but users are encouraged to update the value. We added the following to Section 3.2:*

   *The wind fields include the precision of the winds, which is currently hard-coded as 1 m s⁻¹, which is a rough estimate based on values used previous studies (e.g., Varon et al. 2018, Reuter et al. 2019, Kuhlmann et al. 2021). Users are encouraged to replace the uncertainty with a value suitable for their application and can also use their own wind data from other data sources.*

2. Some discussion on background uncertainty, where appropriate, would also improve the manuscript. S3.3 mentions "uncertainties in the background" but more detail is needed, as this is another important source of error in emission rate estimates. Background uncertainty does not appear to be included in the uncertainty terms at this time, e.g., in S3.4.5, "The uncertainty of the emissions is calculated by propagation of uncertainty from the random uncertainty of the gas columns and the wind speed."

*The background uncertainty depends strongly on trace gas and image resolution (airborne or space-based). We therefore do not include the background uncertainty in more details. To make users aware that the provided uncertainty does not include all sources of errors, we have added the following sentences to Section 3.3.1:*

*"The wind fields include the precision of the winds, which is currently hard-coded as 1 m/s, which is a rough estimate based on values used in previous studies (e.g., Varon et al. 2018, Reuter et al. 2019, Kuhlmann et al. 2021). Users are encouraged to replace the uncertainty with a value suitable for their application and can also use their own wind data from other data sources."*

3. This is certainly beyond the scope of the manuscript and Version 1.0 of ddeq, but it would be great to have support for OCO-2/3 XCO2 data in a future release!

   *We hope to add support for OCO-2 and -3 SAM images relatively soon. In principle, it should work already, but we have not tested this yet. Support for the OCO-2 standard mode will require some more work, because ddeq was designed around the idea of working with wide remote sensing images. For example, ddeq requires that source locations to be within the image, whereas OCO-2 typically observes sources outside the swath.*

Specific comments:

**S2.3.2: "Furthermore, assuming that different trace gases share the same distribution in lateral direction, the method can be expanded to use the standard width and center position estimated for one trace gas directly when fitting the Gaussian function for another gas. This is particularly attractive for the combination of NO2 and CO2 observations..."**

Can the same width always be used when combining NO2 and CO2, or do you need to take into account the NO2 decay as you get further from the source?

*It is possible that $CO_2$ and $NO_2$ plume diverge from each other due to NOX chemistry. For example, NOx lifetime at the edge of the plume is lower because OH concentrations are higher (Krol et al. 2024, https://doi.org/10.5194/egusphere-2023-2519). So far, this effect appears to be too small to contribute significantly to the error budget.*

**S2.5: "Nassar et al., 2022"**

You may also consider citing Hill and Nassar, 2019 (https://www.mdpi.com/2072-4292/11/13/1608)

*We added the reference.*

**S3.4.2: "or when the emission rate is smaller than 0.1 times or larger than 1.9 times the prior expected emission rate"**

What about a scenario where the prior expected emission rate is highly uncertain (e.g., in an undeveloped nation)? Perhaps these limits would be too constraining.

*Yes, this is why we plan to make it possible for the user to choose a suitable range in future as mentioned in Section 3.4.2.*

**S3.4.6 L457: "In the first step, the XCO2"**

Is this just for CO2, or also NO2? It's also the only time XCO2 is mentioned in the manuscript, so you'd need to define it.

*Originally, smoothing and background removal was only for CO2 fields. We have updated the code, making both smoothing and background removal optionally for all gases. The description has been revised accordingly.*

**S3.4.6: "the statistical error of the emission field is also considered."**

Perhaps explain this a bit more.

*We have added the paragraph as follows:*

*"The optimization is first done using the Nelder-Mead method from the scipy library. The uncertainty of the estimated emissions is obtained from the mismatch between emission field E and peak fitting function, i.e. we assume that the chi-square of the fit is the number of degrees of freedom. ddeq also implements the adaptive Metropolis algorithm (Haario et al. 2001) for sampling the posterior distribution (assuming non-informative prior) to obtain an optimized estimate of the fitting parameters and their uncertainty."*

**S4: "As information on emissions from hotspots can be sensitive"**

Please be more specific on what you mean by sensitive here.

*We added "politically sensitive".*

**tutorial-introduction-to-ddeq.ipynb:**

Running the latest release of ddeq (via $pip install ddeq), cell 16 fails with "AttributeError: Module 'scipy' has no attribute 'vectorize'". It looks like era5.py LS00 should have "np.vectorize" instead of "scipy.vectorize". This causes issues in other notebooks too.

*Thank you for testing the code. We have fixed the issue in Version 1.0 in the supplement and the current development version.*

Technical corrections:

L94 "distance along or perpendicular"

*fixed*

Ll08 "**chemically** inert"

*fixed*

L176 "are **the** location"

L176 "the **extent** of the area source"

L259: define "LCSF"

*changed to "general and light cross sectional flux method"*

L275: "location of sources used **is** known"

*fixed*

L277: define "CSV" here and not on L280

*fixed*

Figure 5: the CO2 and NOx emission estimates in the figure do not match those listed in the figure caption.

*fixed*

L308: "GP, CSF and IME method**s**"

*fixed*

L333: "methods"

*fixed*

L360: define "COSMO"

*COSMO stands for "Consortium for Small-scale Modeling". We prefer not to write out acronyms if they do not add any information, but changed this to "COSMO model".*

L390: "to **constrain**"

*fixed*

L394: " random uncertainty"

*fixed*

L477: "in **the** future"

*fixed*

L496: "in **the** future"

*fixed*

L507: define "CORSO"

*Changed to "CO2MVS Research on Supplementary Observations (CORSO) project")*

L509: define "ARES"

*The instrument has been renamed to AVIRIS-4 recently. We slightly modified the sentence.*

L513: "prototype systems for anthropogenic emission monitoring "

*fixed*

**Reviewer 2 (RC2)**

Kuhlmann et al. present an open-source Python library for quantifying point sources of atmospheric trace gases observed by satellites. The library (ddeq) is developed primarily for CO2 and NOx point sources as seen by TROPOMI and the future CO2M mission. It includes tools for data preprocessing (e.g., plume detection and background estimation), emission rate quantification (e.g., CSF, IME, Gaussian plume methods), and data postprocessing (e.g., visualization). The paper is well written and the ddeq library it describes is clearly a valuable contribution to the atmospheric composition remote sensing community. There are several opportunities for continued development of new tools and features, including for other trace gases and satellite missions. I recommend the paper be accepted for publication subject to the minor comments below.

*We like to thank the reviewer for their positive and constructive comments. In the following, we address their comments point by point.*

**Comments**

- L1: It should be clarified that these are atmospheric emissions (as is done in line 13). E.g., "Atmospheric emissions from anthropogenic hotspots…" or "Anthropogenic emissions of air pollutants from hotspots…" or something similar.
  *Thank you. Changed to "Atmospheric emissions from anthropogenic…"*

- L25: It would be more appropriate to describe Sentinel-2 as "multispectral" or "broadband" (vs. "hyperspectral") given its much broader spectral channels than many other instruments.
  *hyperspectral -> multispectral*

- Figure 1: Please mention in the caption what the two different sections of the yellow polygons represent.
  *Changed to "The large yellow polygons delineate the subregions containing the plumes downstream of the source and the smaller polygons show the regions upstream of the source."*

- L87-92: I don't really understand this approach to plume "detection". Doesn't it assume an already detected plume? Otherwise, it would "detect" a plume downwind of any assumed source – even though one might not be detectable in reality.
  *Yes, the second approach implicitly assumes that a plume is located inside the image downwind of the source. Of course, it is it is possible that the plume is not detectable in the image. In this case, the emissions either are below the detection limit (or zero) or the used wind direction is wrong. However, in our benchmark study, we found that the approach, which is used by the LCSF method, works quite well (see Santaren et al. 2024 for details).*

  *The purpose of "plume detection" is the identification of a subregion within the image that is used by the emission quantification algorithm. We have revised the section (new title: "Identification of the plume region"), which hopefully makes the section better understandable.*

- Is the Gaussian plume method still considered lightweight when optimizing the nonlinear (dispersion) parameters?

*The Gaussian plume inversion can get computationally very expensive compared to other lightweight methods when optimizing many non-linear parameters (see Table 1 in Santaren et al. 2024). However, it is still faster than an analytical inversion based on an ensemble of plume-resolving atmospheric transport simulations. The Gaussian plume inversion can also be applied fitting only Q, sigma and V_bg in Eq. 3, which is very fast.*

- L152-153: How are the line integrals at different distances x combined for a polygon extending from x1 to x2? Is the average used?
  *There are several implementations in ddeq. When fitting a Gaussian curve with CSF or LCSF, we do not consider the x-dependency within the polygon. The CSF method includes a "sub-area" method where the polygon is further divided into sub-polygons in the y-direction. To compute the line density, the values within each sub-polygon are averaged before integration, which reduces the sensitivity to clouds (Kuhlmann et al. 2020). Finally, it is also possible to integrate over all values within a polygon and divide by L=x2-x1, resulting in Eq. 17 for the IME approach.*

- L345-346: The 'rs_data' object could use more explanation. I wasn't sure what the 'variable' of that dataset might be.
  *The `rs_data` object is the remote sensing image as described in the second paragraph of Section 3.2. We have revised the paragraph to emphasize that this is about `rs_data`.*

- How is 'rs_data' different from 'datasets', and winds'' from 'wind_folder'?
  *`rs_data` is a single remote sensing image, while datasets provides access to a series of remote sensing images, which is required for the divergence method. Likewise, `wind_folder` makes it possible for the divergence method to read a series `winds` datasets on demand. We have revised the paragraph to better explain the different inputs required for the divergence method.*

- L457: The phrasing here assumes DIV is always applied to XCO2.
  *In the submitted version, the divergence method was only implemented for CO2 and NO2. We have dehardcoded the implementation, which works for other gases now and have rephrased the section to be more general.*

- L491-493: Perhaps remind the reader that some of the plots from the paper were created with ddeq visualization methods – I believe Fig. 1 and 5?
  *We have added this information.*

**Typos and errors**

- L35: I do not understand this sentence, please clarify: "A prototype system of the European CO2MVS capacity is build in CoCO2 project"
  *We have rephrased the paragraph.*

- L52-53: "but they were not included here" is repeated twice in the sentence.
  *fixed*

- L76: "approach" -> "approaches" and "image" -> "images"
  *fixed*

- L176: "extend" -> "extent"
  *fixed*

- L275: sentence is confusing, should it be "used *is* known" ?
  *fixed*

  391: "constraint" -> "constrain"
  *fixed*

- 399: "is" -> "are"
  *fixed*